# Preparation and Characterization of (Al, Fe) Codoped ZnO Films Prepared by Sol–Gel

**Jiangang Wang, Wenjing Shen, Xin Zhang \*, Jianhui Li and Jing Ma**

College of Materials Science and Engineering, Hebei University of Science and Technology,
Shijiazhuang 050000, China; wm094212@163.com (J.W.); shenwenjing0206@163.com (W.S.);
lijianhui_97@163.com (J.L.); majingt@163.com (J.M.)
\* Correspondence: zhxin5210@163.com; Tel.: +86-311-8166-8693

**Abstract:** In this research, Al-doped and (Al, Fe) codoped ZnO films were prepared on glass substrate by the sol–gel method. The surface morphology, structure, and optical property were characterized by scanning electron microscope (SEM), X-ray diffraction (XRD), X-ray photoelectron spectroscopy (XPS), and Ultraviolet-Visible-Near-infrared (UV-Vis-NIR) spectroscopy. The film surface morphologies all exhibited granular characteristics. With the Fe doping concentration increasing, the codoped films had smaller grain size and tended to be smoother. XRD analysis revealed that all films had a hexagonal wurtzite structure. The codoping can contribute to more Al and Fe ions entering the ZnO crystal structure, but result in the crystalline degree of the films decreasing. XPS results showed that the Al and Fe irons in the films exist in the form of trivalent. Moreover, the doped films had higher transmission, especially for codoped (Fe:Al = 3) film, but their absorption edge shifted to the short-wavelength direction.

**Keywords:** ZnO film; sol–gel; (Al, Fe) codoping; structure; optical property

## 1. Introduction

Zinc oxide (ZnO), as an important semiconductor material, has many excellent properties such as wide band gap energy of 3.37 eV, high exciton binding energy of 60 meV, non-toxicity, chemical stability, biocompatibility, and optical transparency. These properties lead to a range of potential applications in ultraviolet photoconductive detectors, solar cells, photocatalysts, and light emitting diodes [1]. Therefore, some researchers have been increasing research efforts to study and even improve ZnO. Among various preparation techniques, the sol–gel method is often used to obtain ZnO films due to its low cost, simple equipment, low temperature operation, easy adjusting composition and dopants, and fabricating large area or complex shape films [2,3]. Moreover, Zn films can be doped with some metal elements to improve their optical, electrical, and magnetic properties [4]. For example, doping of Ag, Cu, and Li elements could allow to obtain p-type ZnO film, which is important for ZnO optoelectronic devices [5–9]. Adding Cd, Mg, Al, and Fe elements might adjust the wide band gap of Zn films, which is an important evaluating index of semiconductor properties [10–13]. Until now, the structural and optical properties of (Al or Fe)-doped Zn films have been reported in detail [14–16]. However, there are few reports about the effect of (Al, Fe) codoping on ZnO films. In this work, (Al, Fe) codoped ZnO films were synthesized by the sol–gel method, and their surface morphology, structural, and optical properties were investigated.

## 2. Materials and Methods

### 2.1. Film Preparation

The undoped, Al-doped, and (Al, Fe) codoped ZnO films were deposited on a glass substrate (CAS: 65997-17-3, Bohui Medical Supplies Co., Ltd, Taizhou, China) by the

dip-coating sol–gel method similar to those described in the literature [14–16]. Zinc acetate 2-hydrate ($Zn(CH_3COO)_2·2H_2O$) (CAS: 5970-45-6, Damao Chemical Reagent Factory, Tianjin, China), ferric nitrate ($Fe(NO_3)_3$) (CAS: 10421-48-4, Yongda Chemical Reagent Factory, Tianjin, China), and aluminium nitrate (($Al(NO_3)_3$)) (CAS: 13473-90-0, Yongda Chemical Reagent Factory, Tianjin, China) were used as precursors, and anhydrous ethanol ($CH_3CH_2OH$) (CAS: 64-17-5, Fuyu Fine Chemical Co., Ltd., Tianjin, China) and triethanolamine ($N(CH_2CH_2OH)_3$) (CAS: 102-71-6, Fuyu Fine Chemical Co., Ltd, Tianjin, China) were used as solvent and stabilizer, respectively. $Zn(CH_3COO)_2·2H_2O$, $Fe(NO_3)_3$ and $Al(NO_3)_3$ were first dissolved in a mixture of $CH_3CH_2OH$ and $N(CH_2CH_2OH)_3$. The exact ratio of the dopant Al and Fe could be estimated for different doping levels. In the mixed solution, the molar ratio of $Zn(CH_3COO)_2·2H_2O$, $CH_3CH_2OH$, and $N(CH_2CH_2OH)_3$ was maintained at 1:100:1. The mixed solution was stirred for 2 h at 60 °C and then aged for 24 h at room temperature. Thereafter, the obtained sol could be used for ZnO film deposition. Before deposition, the glass substrates were washed by acetone, ethanol, and distilled water, respectively, and then dried in clean air. Then, the dip-coating process was carried out in a clean environment. After depositing, the samples were baked in air at 100 °C for 1 h to evaporate the solvent, and then annealed in air at 450 °C for 1 h.

For comparison, an Al doped film was also prepared, with the molar ratios of Zn:Al = 1:0.2. The concentration of (Al, Fe) codoping was varied according to the Zn:Al:Fe molar ratios of 1:0.15:0.05, 1:0.1:0.1, and 1:0.05:0.15. The samples with Al doping and (Al, Fe) codoping (Zn:Al:Fe = 1:0.15:0.0.05, 1:0.1:0.1, and 1:0.05:0.15) were named as Al doped, codoped (Al:Fe = 3), codoped (Al:Fe = 1), and codoped (Fe:Al = 3), respectively. The prepared films all had a thickness of about 350–450 nm.

### 2.2. Film Characterization

The film surface morphologies were observed with a S-4800 scanning electron microscope (SEM) (Hitachi Co., Tokyo, Japan). The film structure was characterized by Smart Lab X-ray diffraction (XRD) (Rigaku Co., Tokyo, Japan), with a Cu K$\alpha$ radiation source (0.15406 nm). The surface chemical state was analyzed with PHI5700 X-ray photoelectron spectroscopy (XPS) (PHI, Inc., Lafayette, LA, USA), using a Mg K$\alpha$ source (1253.6 eV). All binding energies were calibrated with the C 1s peak at 284.5 eV. The absorption spectrum was measured using a UV2600 Ultraviolet-Visible-Near-infrared (UV-Vis-NIR) spectrophotometer, with a resolution of 1 nm.

### 3. Results

### 3.1. Surface Morphology

The SEM photographs of ZnO, Al doped, and codoped (Al:Fe = 1) films with low magnification are presented in Figure 1. These films all show a homogeneous and dense morphology and have no obvious defect such as cracks and holes on the surfaces. The surface morphologies of other films are similar to those shown in Figure 1.

The SEM photographs of different ZnO films with high magnification are shown in Figure 2. It can be seen that all films present a granular morphology. Compared with the ZnO film (Figure 2a), the doping does not affect the surface morphological characteristic (Figure 2b), but the grain size and surface roughness changes obviously. In Figure 2c, the surface of Al doped film tends to be smooth. After (Al, Fe) codoping, with the Fe doping concentration increasing, the gain size becomes smaller and smaller, and simultaneously, the films tend to be smoother, as in Figure 2d–f. These results suggest that (Al, Fe) codoping can allow to obtain the films with high quality.

### 3.2. XRD Analysis

In order to study the effect of (Al, Fe) doping on the microstructure of ZnO films, the XRD analysis of different samples were carried out, and the result is shown in Figure 3. The XRD patterns have been treated to remove the substrate background. The diffraction peaks in Figure 3 all match the standard diffraction pattern of the ZnO crystal (JCPDS 36-1451),

which confirms the hexagonal wurtzite structure [17]. However, the XRD patterns do not show the phenomenon of preferential orientation along the c-axis ((002) plane) reported in the literature [18]. Ohyama et al. pointed that the film thickness and annealing treatment could greatly affect the growth orientation of the ZnO film [19,20]. Except the diffraction peaks of ZnO, no peaks relating to Fe and Al or its oxides can be observed in Figure 3, which means that doping does not change the film structure.

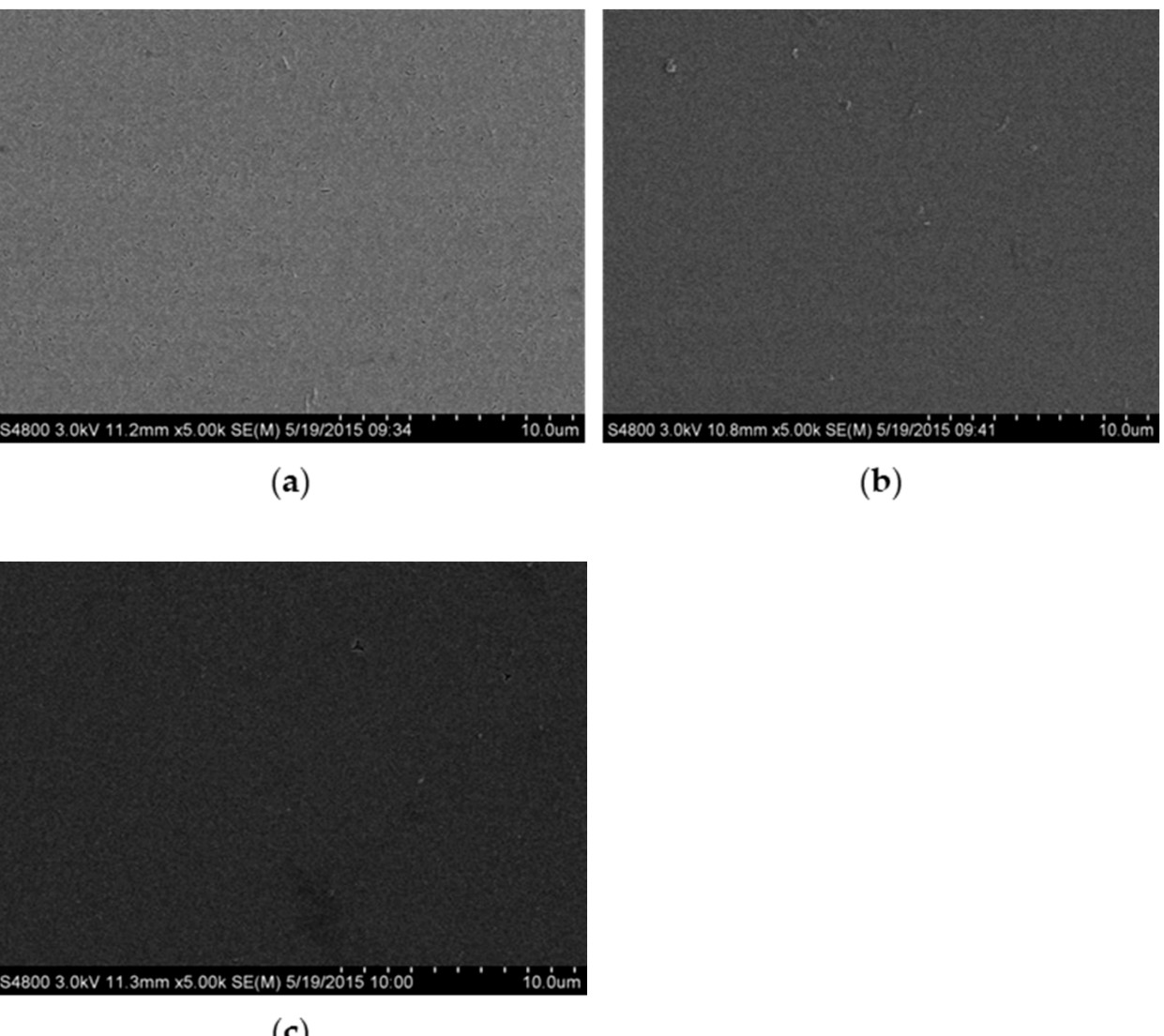

**Figure 1.** SEM photographs of different ZnO films ((**a**) ZnO, (**b**) Al doped, (**c**) Codoped (Al:Fe = 1)).

However, it can be found in Figure 3 that some diffraction peaks are moved to higher angles for the doped films. For example, the 2θ value of (002) peaks is shifted from 34.376° for pure ZnO film to 34.552°, 34.551°, 34.552°, and 34.677° for Al doped, codoped (Al:Fe = 3), codoped (Al:Fe = 1) and codoped (Fe:Al = 3). These shifts imply that Fe and Al irons have been incorporated into the ZnO lattice, i.e., substitution of Fe and Al irons on Zn irons. According to the Bragg equation ($d = \lambda/2\sin\theta$, where $\lambda$ is the wavelength of X-ray radiation source, $\theta$ is the Bragg diffraction angle), since the diffraction peaks shift to higher angles, the corresponding interplanar spacing will decrease, as shown in Table 1. This is because the ionic radius of $Fe^{3+}$ (0.068 nm) and $Al^{3+}$ (0.051 nm) is smaller than that of $Zn^{2+}$ (0.074 nm), and then the substitution of Fe and Al ions in the ZnO lattice can decrease the values of the lattice constant, namely, the volume shrinkage of the ZnO crystal structure.

This result is in agreement with that of other sol–gel-deposited doped ZnO films [14,16]. It can be known from the above analysis that with the Fe doping concentration increasing, the (002) peak of codoped films has a larger 2θ value, even exceeding that of Al doped film. Therefore, the codoping results in more Fe and Al ions entering the ZnO crystal structure.

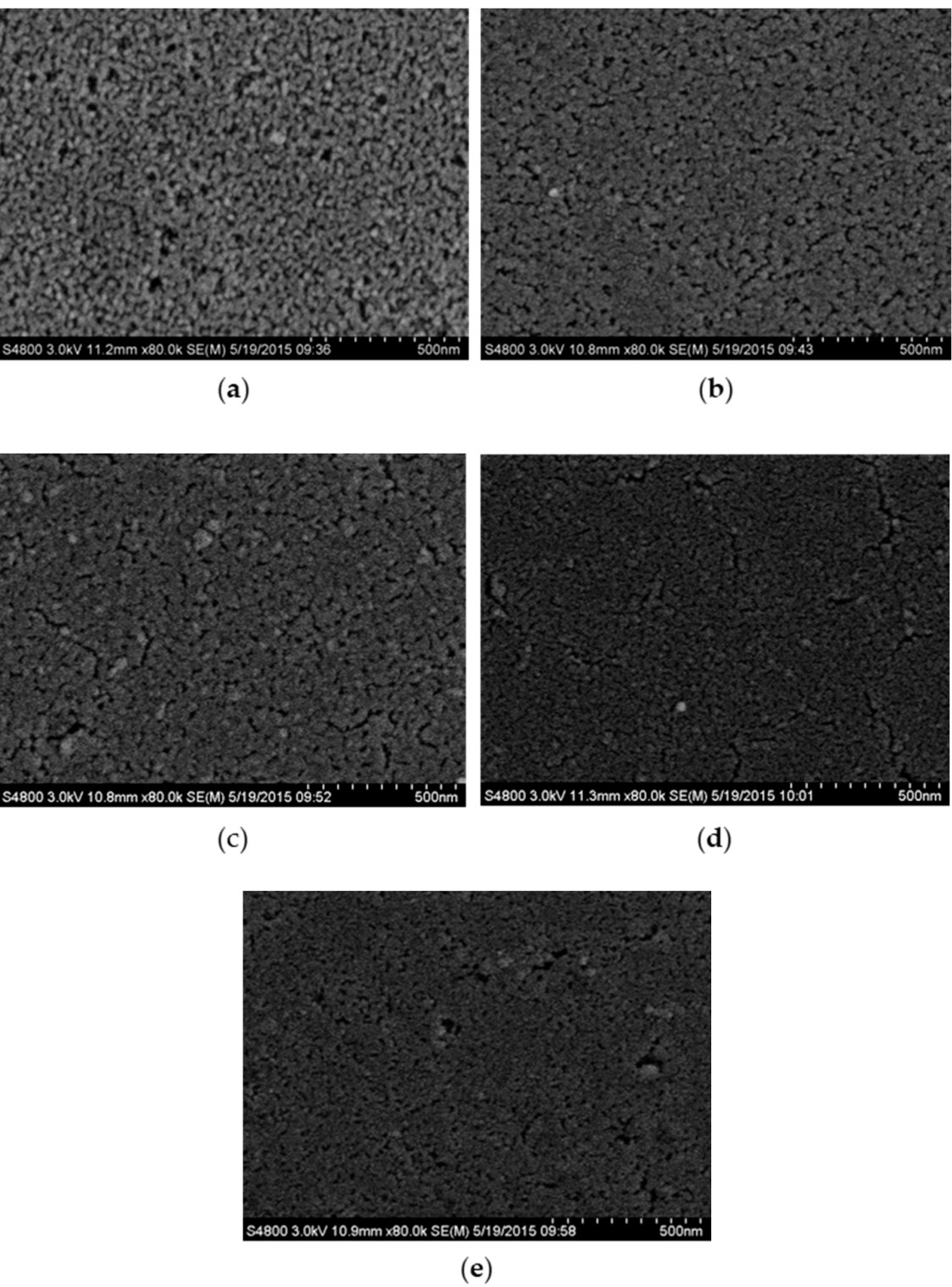

**Figure 2.** SEM photographs of different ZnO films ((**a**) ZnO, (**b**) Al doped, (**c**) Codoped (Al:Fe = 3), (**d**) Codoped (Al:Fe = 1), (**e**) Codoped (Fe:Al = 3)).

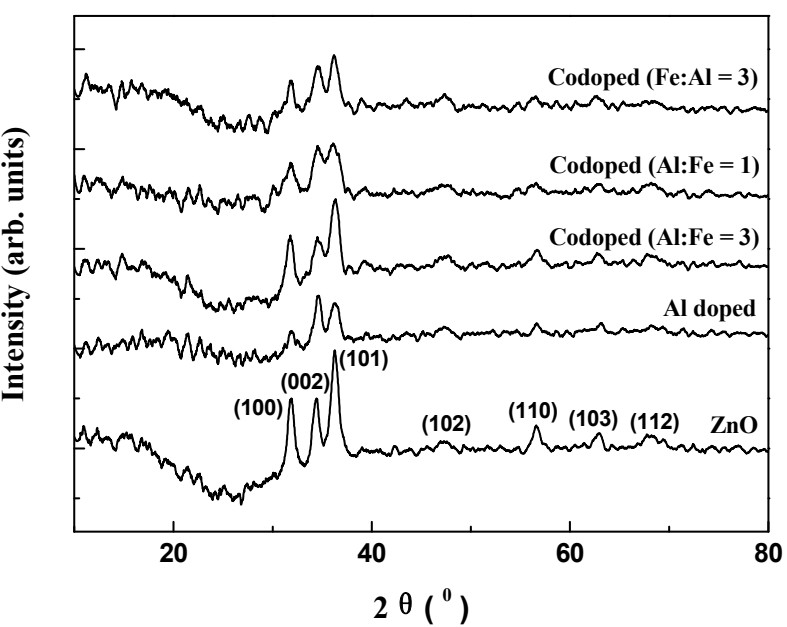

**Figure 3.** XRD patterns of different ZnO films.

**Table 1.** The 2θ value, interplanar spacing, and grain size of (002) peak of different films.

| Films | ZnO | Al Doped | Codoped (Al:Fe = 3) | Codoped (Al:Fe = 1) | Codoped (Fe:Al = 3) |
|---|---|---|---|---|---|
| 2θ value | 34.376° | 34.552° | 34.551° | 34.552° | 34.677° |
| interplanar spacing | 0.2607 nm | 0.2594 nm | 0.2594 nm | 0.2594 nm | 0.2585 nm |
| grain size | 8.984 nm | 8.544 nm | 6.335 nm | 6.239 nm | 5.856 nm |

On the other hand, after doping Al or Fe element in ZnO film, the relative intensities of diffraction peaks decrease, which indicates that the crystalline degree has come down in the doped film, especially for codoped (Fe:Al = 3) films. This might be due to the lattice disorder and strain induced by the substitution of Fe and Al ions for Zn ions. A similar result has been reported for the reduction of the crystalline degree in the doped ZnO films [13,16]. Moreover, it can also be measured from Figure 3 that the full widths at half-maximum (FWHMs) of diffraction peaks for ZnO films are decreased after doping. The grain size (D) of the films could be calculated by the Scherrer relation: $D = 0.89\lambda/\beta\cos\theta$ (where $\lambda$ is the wavelength of X-ray radiation source, $\theta$ is the Bragg diffraction angle, and $\beta$ is the FWHM of diffraction peak), as shown in Table 1. Therefore, the doping decreases the grain size for the films. This size difference may be attributed to the stress arising from the difference between the ion radii of dopant and Zn, which affects the normal growth of the ZnO grain. This is consistent with the results reported by Chen et al. [21] and Wang et al. [22]. As the result of the reduction of crystalline degree and grain size, the codoped films have a smoother surface, especially with higher Fe doping content, as shown in Figures 1 and 2.

### 3.3. XPS Analysis

For characterizing the chemical states of (Al, Fe) codoped ZnO films, codoped (Al:Fe = 1) film was measured by XPS. Figure 4 gives the XPS full spectrum of codoped (Al:Fe = 1) film. The photoelectron peaks of Zn, O, and C elements can be clearly observed in Figure 4. The C peak is formed due to the contamination and absorption of the sample surface. However, there are no obvious peaks of Fe and Al elements at the corresponding positions. This is because the doping amount is small and thus the dopants have no evident

effect on the crystal structure. Table 2 lists the atomic content of each element in codoped (Al:Fe = 1) film.

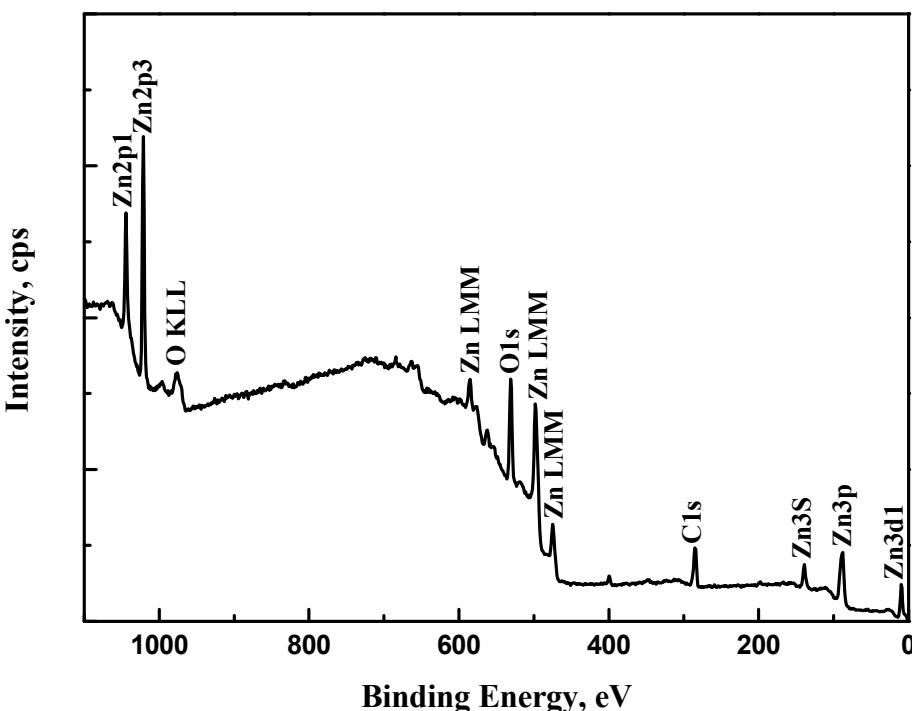

**Figure 4.** XPS full spectrum of codoped (Al:Fe = 1) film.

**Table 2.** Atomic content of each element in codoped (Al:Fe = 1) film.

| Element | C 1s | O 1s | Al 2p | Fe 2p3 | Zn 2p3 |
|---|---|---|---|---|---|
| Content (%) | 41.42 | 40.30 | 1.04 | 0.44 | 16.80 |

Figure 5 shows the high resolution spectra of Zn 2p3, O 1s, Al 2p, and Fe 2p3 peaks in XPS full spectrum of codoped (Al:Fe = 1) film. The Zn 2p3 peak at the binding energy of 1021.4 eV is consistent with that in ZnO reported in the literature [23], which proves that ZnO keeps the hexagonal crystal structure. For Al 2p spectra, there is only one peak at 73.9 eV, while the Fe 2p3 peak at 710.6 eV is relatively weak. Compared with the standard peaks of Al 2p (74.3 eV) and Fe 2p3 (710.7 eV), both peaks exhibit a certain shift, which is attributed to the change of electronic valence state due to Al and Fe irons entering into the ZnO crystal lattice. The O 1s spectra consist of three peaks: one at 530.1 eV for Zn–O–Zn bonds [24], one at 531.4 eV from (Fe or Al)–O–Zn bonds after doping, and another one at 532.3 eV for –OH groups [25]. The above results prove that the Al and Fe elements are successfully doped into the ZnO lattice, and both elements existed in the form of trivalent. However, the Al and Fe doping does not change the hexagonal crystal structure of ZnO.

### 3.4. Optical Property

Figure 6 displays the transmittance spectra of different ZnO films. It can be observed that the transmission of all films is higher than 80% in the visible region of 400–800 nm. Compared with the ZnO film, the doped films have higher transmission, and the transmission of codoped (Al:Fe = 3) film can even reach up to about 90%. However, the transmission of codoped (Al:Fe = 1) and codoped (Fe:Al = 3) films is slightly lower that of Al doped films. Moreover, all films shows a strong absorption in the ultraviolet region of 300–400 nm, which is attributed to the intrinsic absorption originated from the direct transition of electrons [26]. It can also be noted that the absorption edge of the films shifts to the short-wavelength

direction after doped, which is in agreement with that reported in the literature [27,28]. This might be attributed to the reduction of surface roughness [29], which is accordant with the result of SEM observation. According to the formula of Eg = hc/λg, the band gap energy (Eg) of the films can be calculated to be 3.64 eV, 3.99 eV, 3.82 eV, 3.95 eV, and 4.00 eV for ZnO, Al doped, codoped (Al:Fe = 3), codoped (Al:Fe = 1), and codoped (Fe:Al = 3), respectively. Therefore, the optical band-gap ($E_g$) of the films after doped will be increased due to the shift of the absorption edge. This phenomenon further confirms the Fe or Al substitution in the ZnO structure. The change of the band gap energy of ZnO film can be explained by the energy band theory. When Fe or Al irons go into the crystal lattice of Zn to replace the Zn irons, they will provide additional free carriers [30]. The increased carriers firstly occupy the low level in the conduction band. When the ultraviolet light irradiates, the electrons in the valence band can enter the conduction band over the forbidden band. However, the low level in the conduction band has been filled, thus the electrons only jump to the high level, which corresponds to the forbidden band broadened. The higher $E_g$ could result in the increase of the open circuit voltage, so the maximum work temperature of materials may be increased.

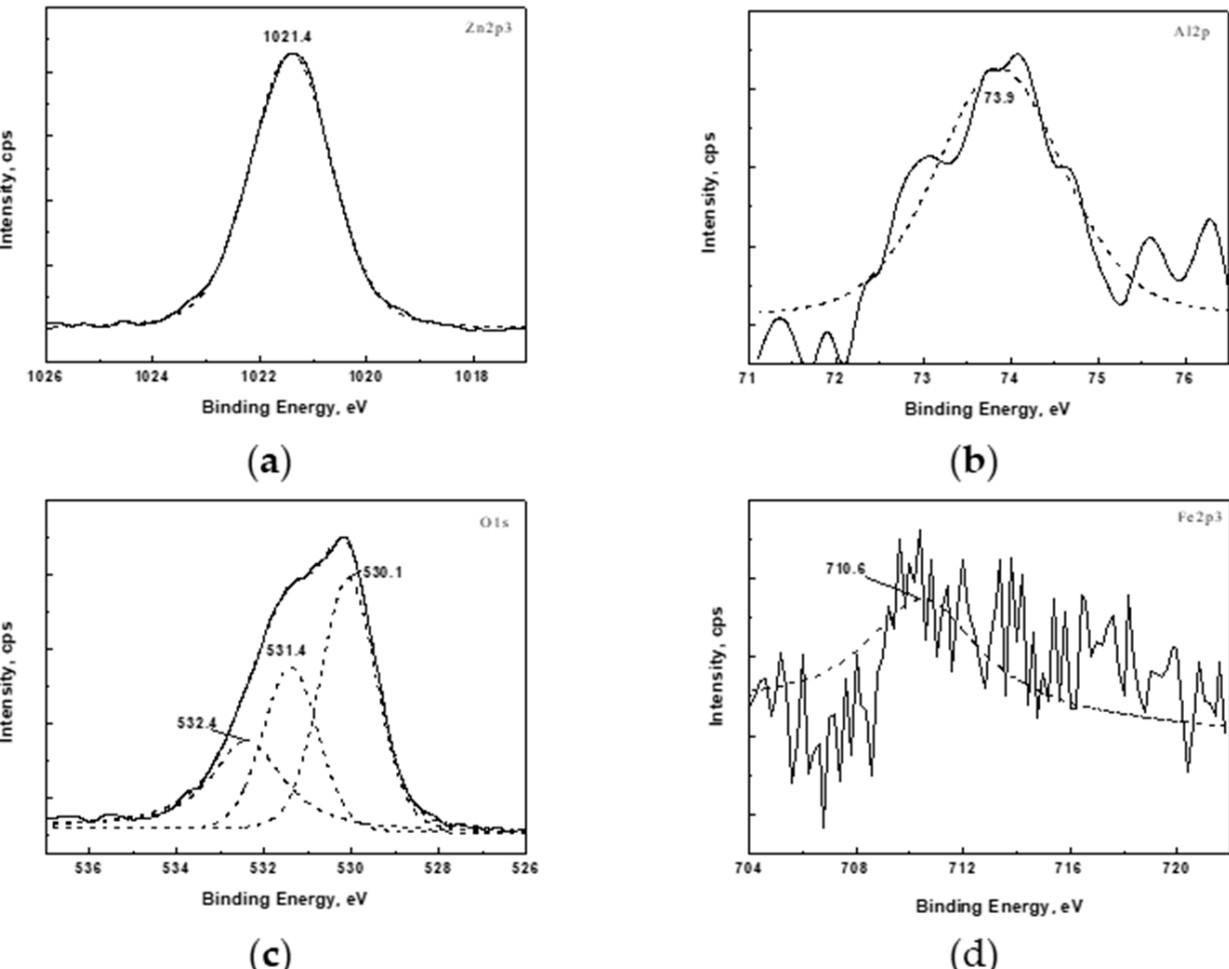

**Figure 5.** XPS spectra of Zn 2p3, O 1s, Al 2p, and Fe 2p3 peaks in codoped (Al:Fe = 1) film. (**a**) Zn 2p3, (**b**) O 1s, (**c**) Al 2p, (**d**) Fe 2p3.

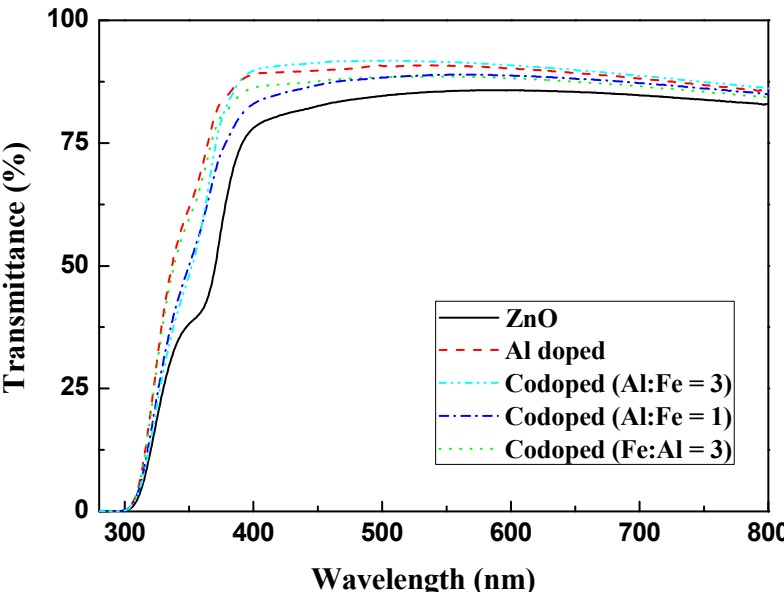

**Figure 6.** Transmittance spectra of different ZnO films.

### 4. Conclusions

　　ZnO film, as an important semiconductor material, has often improved its properties by doping. However, there are few reports about the effect of (Al, Fe) codoping on ZnO films. In this paper, (Al, Fe) codoped ZnO films were prepared by the sol–gel method and were researched in detail through surface morphology, structure, and optical properties. The prepared films all exhibited a granular morphology and had a hexagonal wurtzite structure. Comparatively, (Al, Fe) codoped films became smother and the grain size became smaller, especially for codoped (Fe:Al = 3) films with higher Fe doping concentration. The analysis results indicate that Al and Fe doping does not change the film structure. The Al and Fe irons in the films exist in the form of trivalent. However, the substitution of Fe and Al irons on Zn irons causes the diffraction peaks to move to a lower angle value. Moreover, compared with the ZnO film, all doped films have a higher transmission in the region of 400–800 nm due to the reduction of surface roughness, but their absorption edge shifts to the short-wavelength direction. These changes can increase the operating temperature in the actual application of the ZnO film.

**Author Contributions:** Conceptualization, J.W., X.Z. and W.S.; Methodology, J.L. and J.M.; Formal analysis, X.Z. and W.S.; Investigation, X.Z., W.S. and J.W.; Resources, W.S. and X.Z.; Data curation, W.S.; Writing—original draft preparation, X.Z. and W.S.; Writing—review and editing, X.Z., W.S. and J.W.; Visualization, J.M. and J.L.; Supervision, J.M.; Project administration, J.W.; Funding acquisition, J.W. All authors have read and agreed to the published version of the manuscript.

**Funding:** This work was funded by the Military civilian science and Technology Collaborative Innovation Project, Grant Nos 20351801D, S&T Program of Hebei, Grant Nos 20351801D and 20564401D and Key projects of Hebei Provincial Department of Education, Grant Nos ZD2020189.

**Institutional Review Board Statement:** Not applicable.

**Informed Consent Statement:** Not applicable.

**Data Availability Statement:** Data sharing is not applicable to this article.

**Conflicts of Interest:** The authors declare no conflict of interest.

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
