# Peer review of "Preparation and Characterization of (Al, Fe) Codoped ZnO Films Prepared by Sol–Gel"

_coatings, doi:10.3390/coatings11080946_

Round 1

Reviewer 1 Report

This manuscript describes the preparation of Al-doped and (Al, Fe) codoped ZnO films on glass substrate by sol-gel method. The surface morphology, structure and optical property were characterized by scanning electron microscope (SEM), X-ray diffraction (XRD), X-ray photoelectron spec-troscopy (XPS) and UV-Vis-NIR spectrophotometer. With the Fe doping concentration increasing, the codoped films have smaller grain size and tend to be smoother. XRD analysis revealed that all films have a hexagonal wurtzite structure. The codoping can contribute to more Al an Fe ions entering the ZnO crystal structure, but result in the crystalline degree of the films decreasing. XPS results showed that the Al and Fe irons in the films exist in the form of trivalent. Moreover, the dopped films have higher transmission, especially for ZAF3 film, but their absorption edge shifts to short-wavelength direction.

However, the manuscript shows different critical points:

  • Firstly, the authors must improve the introduction such as the conclusion because they appear short, and not contextualized this work whit recently literature. For this reason, the manuscript appears unclear and devoid of novelty. The authors must be compare their experimental result and could improve the description of the possible application of these materials.
  • The authors could improve the morphological characterization to added information about the thickness of films on glass.
  • The authors must verify the writing of “HOCH2CH2)3N” at different point of manuscript.

For all these considerations, this manuscript cannot be accepted publication but after this minor revision.

Author Response

  1. We have modified the conclusion according to the reviewer opinion.
  2. We have given the thickness of films in the manuscript.
  3. We have unified its writing of “HOCH2CH2)3N” in the manuscript.

Reviewer 2 Report

The authors present the preparation and characterization of (Al, Fe) codoped ZnO films fabricated via sol-gel method. There are many technical and structural errors in the manuscript that need serious attention. I believe the results and the research idea don't have potential to be published at this stage. Hence, I would like the reject the paper publication in coatings based on the following comments.

1- The authors choose a very old research topic that demands very high novelty in terms of research idea and exceptional results. The paper lacks both. The research idea is outdated and there was nothing novel in the research.

2- There are many structural and grammatical errors in the manuscript. It should be checked with a linguistic expert. Also, there are many typos in the manuscript.

3- The introduction is not applicable. It lacks all the necessary ingredients required for a professional introduction section.

4- Provide an analytical comparison of why the authors choose dip coating growth method over other methods of growth such as ALD, MOCVD, MBE, Hydrothermal and Microwave assisted method?

5- Provide CAS number of all the chemicals and substrates in the experimental section. Also, provide the doping %?

6- It looks like the paper is a factual description of results rather than an analytical and scientific exploration of new phenomenon. 

Author Response

  1. In this manuscript, it has been pointed out that ZnO films can be doped with some metal elements to improve their properties, but there are few reports about the effect of (Al, Fe) codoping on ZnO films, so we detailedly studied them in this manuscript. These results can provide a basis of the selection of doping elements for improving the ZnO film properties and the applications.
  2. We have carefully examined the structure and grammatical errors in the writings and corrected them.
  3. The introduction points that zinc oxide films can improve their performance by doping and shows some common doping elements. This manuscript is mainly the study of (Al, Fe) codoped zinc oxide films due to their few reports. The manuscript also compares the results with those in some literatures and explains the morphology, structure and performance of the codoped film.
  4. Due to its low cost, simple equipment, low temperature operation, easy adjusting composition and dopants, and fabricating large area or complex shape films, we prepared the ZnO films by sol-gel method.
  5. We have already provided CAS number of all the chemicals and substrates and the doping % in the experimental section.
  6. I think that perhaps the results in this manuscript can provide a basis of the selection of doping elements for improving the ZnO film properties and the applications.

Reviewer 3 Report

In this work, Al-doped and (Al, Fe) co-doped ZnO films were prepared on glass substrate by sol-gel method. The surface morphology, structure and optical property were characterized by scanning electron microscope (SEM), X-ray diffraction (XRD), X-ray photoelectron spectroscopy (XPS) and UV-Vis-NIR spectroscpy.

There are several major issues that need to be resolved.

Unless the authors analyze further their work and compare with the literature, this work cannot be published.

1.First of all some syntax and grammar errors should be corrected.

Starting from the abstract, the authors state .. The surface morphology, structure and optical property were character-ized by scanning electron microscope (SEM), X-ray diffraction (XRD), X-ray photoelectron spec-troscopy (XPS) and UV-Vis-NIR spectrophotometer.."

it should be ..spectroscopy, not spectrophotometer.

2.I am afraid the introduction is not efficient. I think there is some information missing, for example what is the novelty of this work compared to the literature.

3.There are some references missing in the "Film preparation" section. The authors present a well known experimental procedure for the synthesis of ZnO films using the sol-gel approach.

4.in the "XRD analysis" section, the authors use the term XRD "spectra".

XRD is not a spectrum, the authors should use the term "pattern" instead.

5.Figure 3 presents XRD patterns of different ZnO films. I suggest the authors to comment the diffraction peaks of their samples. If one follows the experimental procedure of the literature, he will get just the (002) direction at 34.42 degrees. I kindly ask the authors to comment on the crystalinity of their samples; how come the get all the diffraction peaks and not just the one perpendicular on the substrate (002)?

6.Taking into account the Optical properties of their samples, could the authors calculate the energy gap of their samples and comment on that?

They could easily calculate from the spectra, if they know the thickness of their samples.

This work presents the synthesis of Al-doped and (Al, Fe) co-doped ZnO thin films  on glass substrate by sol-gel process. This is not novel at all.

Unless the authors analyze further their work and compare with the literature, this work cannot be published.

Author Response

  1. We have replaced “UV-Vis-NIR spectrophotometer “ with “UV-Vis-NIR spectroscopy” in the abstract.We carefully examined the errors in the manuscript and corrected them.
  2. In this manuscript, it has been pointed out that ZnO films can be doped with some metal elements to improve their properties, but there are few reports about the effect of (Al, Fe) codoping on ZnO films, so we detailedly studied them in this manuscript. These results can provide a basis of the selection of doping elements for improving the ZnO film properties and the applications.
  3. We have supplemented the references for the preparation of ZnO films by the sol-gel approach.
  4. We have modified the corresponding places in the manuscript.
  5. In our manuscript, we can get all the diffraction peaks in XRD patterns when the prepared films with the thickness of 350~450nm are treated at 450ºC for 1h. Taking the more studied (002) diffraction peaks as an example, we analyze the effect of codoping on the ZnO film structure.
  6. We have already calculated the energy gap of the film samples and comment on that. 

Reviewer 4 Report

Your manuscript entitled “Preparation and characterization of (Al, Fe) codoped ZnO films prepared by sol gel” is interesting and relates with publication scope of “Coatings”. However, I fell that the writing quality is quite poor and therefore you need to majorly revise this manuscript prior the acceptance in “Coatings”.

Some amendments are following:

  1. On page 2, paragraph 2, line 4, the nomenclatures for the sample names are unclear (ZA, ZAF3, ZAF1 and ZFA3). ZAF3 and ZFA3 are causing confusion. Please rename them.
  2. The SEM pictures in Figures 1 and 2 are not clear and causing misunderstanding. Why they are two zooms in Figures 1 and 2? It is better to merge them into 1 with notation from (a) to (h). The quality of Figures 2c and 2d are terrible. It is better to increase the resolution of both figures. For the scale bars, they are not clear. Please replace them with white bars with the same sizes.
  3. Figure 3 of the XRD pattern from different ZnO films. Why the doped ones worse that undoped film? Then this is this consistent with the improvements of the transmittances for doped ZnO films in Figure 6? Please add discussion in the manuscript.
  4. Table 1 on page 5, there is a vertical word “S u p p”, what is it?
  5. For atomic contents on Table 2, why it is only ZAF1? Did you compare with other samples? Please add the atomic contents for other samples.
  6. Please be careful with the errors in the writings, e.g. page 2, paragraph 2 line 3, “…with Al dopingand”.

Author Response

  1. We have replaced Figure 2c and 2d with clearer pictures. Because the prepared films have similar appearance features at low multiples, the surface morphology of the three films is only given in Figure 1 to indicate the film appearance features. Now the scale bars in Figures are clear.
  2. We have already added discussion in the manuscript about XRD result and transmittances. After doping, the crystalline degree of the doped films has come down, especially for codoped (Fe/Al=3) films. This might be due to the lattice disorder and strain induced by the substitution of Fe and Al ions for Zn ions.
  3. Maybe due to the page showing, we didn't find the word “S u p p”.
  4. In order to characterize the chemical states of (Al, Fe) codoped ZnO films, we only analyze one codoped sample. Moreover, some references have also proved that the dopant content has no or less effect on the chemical states of atoms.
  5. We carefully examined the errors in the writings and corrected them.

Round 2

Reviewer 2 Report

I am not satisfied with the novelty of the study. Secondly, it is not confirm which application are to be considered for codoping. The idea does not add anything novel to science and won't be a good addition to Costings. Hence, I retain my decision to reject the paper publication in Coatings.

Reviewer 3 Report

The authors have revised their manuscript following most of the suggested revisions.

This manuscript could be published in its present form.

Reviewer 4 Report

The revision is Ok,